# GraphLog: A Benchmark for Measuring Logical Generalization in Graph Neural Networks

## Abstract

Relational inductive biases have a key role in building learning agents that can generalize and reason in a compositional manner. While relational learning algorithms such as graph neural networks (GNNs) show promise, we do not understand their effectiveness to adapt to new tasks. In this work, we study the *logical generalization* capabilities of GNNs by designing a benchmark suite grounded in first-order logic. Our benchmark suite, GraphLog, requires that learning algorithms perform rule induction in different synthetic logics, represented as knowledge graphs. GraphLog consists of relation prediction tasks on 57 distinct procedurally generated logical worlds. We use GraphLog to evaluate GNNs in three different setups: single-task supervised learning, multi-task (with pre-training), and continual learning. Unlike previous benchmarks, GraphLog enables us to precisely control the logical relationship between the different worlds by controlling the underlying first-order logic rules. We find that models' ability to generalize and adapt strongly correlates to the availability of diverse sets of logical rules during multi-task training. We also find the severe catastrophic forgetting effect in continual learning scenarios, and GraphLog provides a precise mechanism to control the distribution shift. Overall, our results highlight new challenges for the design of GNN models, opening up an exciting area of research in generalization using graph-structured data.

## 1 Introduction

Relational reasoning, or the ability to reason about the relationship between objects and entities, is considered one of the fundamental aspects of intelligence (Krawczyk et al., 2011; Halford et al., 2010), and is known to play a critical role in cognitive growth of children (Son et al., 2011; Farrington-Flint et al., 2007; Richland et al., 2010). This ability to infer relations between objects/entities/situations, and to compose relations into higher-order relations, is one of the reasons why humans quickly learn how to solve new tasks (Holyoak and Morrison, 2012; Alexander, 2016). The perceived importance of relational reasoning for generalization has fueled the development of several neural network architectures that incorporate relational inductive biases (Battaglia et al., 2016; Santoro et al., 2017; Battaglia et al., 2018). Graph neural networks (GNNs), in particular, have emerged as a dominant computational paradigm within this growing area (Scarselli et al., 2008; Hamilton et al., 2017a; Gilmer et al., 2017; Schlichtkrull et al., 2018; Du et al., 2019). However, despite the growing interest in GNNs and their promise of relational generalization, we currently lack an understanding of how *effectively* these models can adapt and generalize across distinct tasks.

In this work, we study the task of *logical generalization* in the context of relational reasoning using GNNs. One example of such a reasoning task (from everyday life) can be in the context of a family-graph where the nodes are family members, and edges represent the relationships (brother, father, etc). The objective is to learn logical rules, which are compositions of a specific form, such as "the *son* of my *son* is my *grandson*". As new compositions of existing relations are discovered during the lifetime of a learner, (e.g., the *son* of my *daughter* is my *grandson*), it should remember the old compositions, even as it learns new compositions (just like we humans do). This simplistic example can be extended to the more complex (and yet practical) scenarios like identifying novel chemical compositions, or recommender systems, where agents have to learn and retain compositions of existing relations.

| Dataset | IR | D | CG | M | S | Me | Mu | CL |
|---|---|---|---|---|---|---|---|---|
| CLEVR (Johnson et al., 2017) | ✓ | ✗ | ✗ | Vision | ✓ | ✗ | ✗ | ✗ |
| CoGenT (Johnson et al., 2017) | ✓ | ✗ | ✓ | Vision | ✓ | ✗ | ✗ | ✗ |
| CLUTRR (Sinha et al., 2019) | ✓ | ✗ | ✓ | Text | ✓ | ✗ | ✗ | ✗ |
| SCAN (Lake and Baroni, 2017) | ✓ | ✗ | ✓ | Text | ✓ | ✓ | ✗ | ✗ |
| SQoOP (Bahdanau et al., 2018) | ✓ | ✗ | ✓ | Vision | ✓ | ✗ | ✗ | ✗ |
| TextWorld (Côté et al., 2018) | ✗ | ✓ | ✓ | Text | ✓ | ✓ | ✓ | ✓ |
| GraphLog (Proposed) | ✓ | ✓ | ✓ | Graph | ✓ | ✓ | ✓ | ✓ |

Table 1: Features of related datasets that: 1) test compositional generalization and reasoning, and 2) are procedurally gnerated. We compare the datasets along the following axis: Inspectable Rules (**IR**), **D**iversity, Compositional Generalization (**CG**), **M**odality and if the following training setups are supported: **S**upervised, **Me**ta-learning, **Mu**ltitask & Continual learning (**CL**).

| | |
|---|---|
| Number of relations | 20 |
| Total number of *WorldGraph*s | 57 |
| Total number of unique rules | 76 |
| Training Query graphs per *WorldGraph* | 5000 |
| Validation Query graphs per *WorldGraph* | 1000 |
| Testing Query graphs per *WorldGraph* | 1000 |
| Average number of nodes per Query graph | 14.55 |
| Average number of edges per Query graph | 17.94 |
| Average number of triangles per Query graph | 11.27 |
| Number of rules per *WorldGraph* | 20 |
| Maximum length of resolution path | 10 |
| Minimum length of resolution path | 2 |

Table 2: Aggregate statistics of the worlds in GraphLog. Statistics for each individual world are in the Appendix.

We study the effect of such generalization by analyzing the ability of GNNs in learning new relation compositions, leveraging first-order logic. In particular, we study how GNNs can induce logical rules and generalize by combining such rules in novel ways after training. We propose a benchmark suite, GraphLog, that is grounded in first-order logic. Figure 1 shows the setup of the benchmark. Given a set of logical rules, we create a diverse set of logical *worlds* with a different subset of rules. For each world (say $W_k$), we sample multiple knowledge graphs (say $g_i^k$). The learning agent should learn to induce the logical rules for predicting the missing facts in these knowledge graphs.

Using our benchmark, we evaluate the generalization capabilities of GNNs in the supervised setting by predicting inductively unseen combinations of known rules within a specific logical *world*. We further analyze how various GNN architectures perform in the multi-task and continual learning scenarios, where they have to learn over a set of logical worlds with different underlying logics. Our setup allows us to control the *distribution shift* by controlling the similarity between the different *worlds*, in terms of the overlap in logical rules between different *worlds*.

Our analysis provides the following insights about logical generalization capabilities of GNNs:

- Two architecture choices for GNNs strongly (and positively) impact generalization: **1)** incorporating multi-relational edge features using attention, and **2)** modularising GNN architecture to include a parametric *representation function*, to learn representations for the relations using a knowledge graph structure.

- In the multi-task setting, training a model on a more diverse set of logical *worlds* improves generalization and adaptation performance.

- All the evaluated models exhibit catastrophic forgetting. This indicates that the models are not learning transferable representations and compositions and just overfitting to the current task —highlighting the challenge of lifelong learning in context of logical generalization and GNNs.

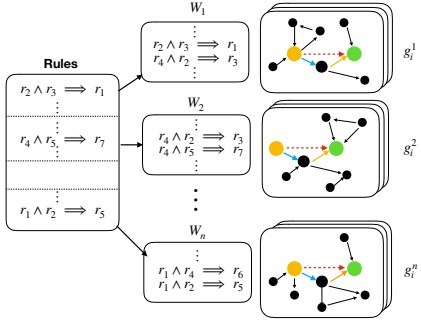

Figure 1: GraphLog setup: A set of rules (grounded in propositional logic) is partitioned into overlapping subsets. It is used to define unique *worlds*. Within each *world* $W_k$, several knowledge graphs $g_i^k$ (governed by rule set of $W_k$) are generated.

## 2 BACKGROUND AND RELATED WORK

**GNNs**. Several GNN architectures have been proposed to learn representations of the graph inputs (Scarselli et al., 2008; Duvenaud et al., 2015; Defferrard et al., 2016; Kipf and Welling, 2016; Gilmer et al., 2017; Hamilton et al., 2017b; Schlichtkrull et al., 2018). Previous works have focused on evaluating GNNs in terms of their expressive power (Barceló et al., 2019; Morris et al., 2019; Xu et al., 2018), usefulness of features (Chen et al., 2019), and explaining their predictions (Ying et al., 2019). Complementing these works, we evaluate GNN models on the task of logical generalization.

**Knowledge graph completion**. Many knowledge graph datasets are available for relation prediction tasks (also known as knowledge base completion). Prominent examples include Freebase15K (Bordes et al., 2013), WordNet (Miller, 1995), NELL (Mitchell and Fredkin, 2014), and YAGO (Suchanek et al., 2007; Hoffart et al., 2011; Mahdisoltani et al., 2013). Since these datasets are derived from real-world knowledge graphs, they are generally noisy and incomplete, and many facts are not available in the underlying knowledge bases (West et al., 2014; Paulheim, 2017). Moreover, the underlying logical rules are often opaque and implicit (Guo et al., 2016), thus reducing the utility of these datasets for understanding the logical generalization capability of neural networks.

**Procedurally generated datasets for reasoning**. Several procedurally generated benchmarks have been proposed to study the relational reasoning and compositional generalization properties of neural networks (Table 1). These datasets provide a controlled testbed for evaluating compositional reasoning capabilities of neural networks in isolation. Based on these insightful works, we enumerate the four key desiderata that, we believe, such a benchmark should provide:

1. Human **interpretable rules** should be used to generate the dataset.
2. The datasets should be **diverse**, and the compositional rules used to solve different tasks should be distinct, so that adaptation on a new task is not trivial. The degree of similarity across the tasks should be configurable to evaluate the role of diversity in generalization.
3. The benchmark should test for **compositional generalization**, which is the ability to solve unseen combinations of existing rules, thus generalizing through the composition of known concepts.
4. The benchmark should support creating a **large number of tasks** and enable a more fine-grained inspection of the generalization capabilities of the model in different setups, e.g., supervised learning, multi-task learning, and continual learning.

As shown in Table 1, GraphLog is unique in satisfying all of these desiderata. We highlight that unlike previous works which have been largely focused on the image and text modalities, GraphLog is one of the unique attempts to test logical generalization on graph-structured data using Graph Neural Networks. CLUTRR (Sinha et al., 2019) also provides similar underlying graphs in the text corpus, although GraphLog is different based on the following properties:

- CLUTRR consists of a single rule world which in turn contains only 15 rules. GraphLog extends the concept to a more general framework where a user can define how many rules can occur in a single world, as well as define multiple such worlds.

- GraphLog allows building multiple worlds consisting of either the same, overlapping, or distinct set of rules - which allows practitioners to test multi-task and continual learning scenarios in minute detail by controlling the distribution shift, which is a key difference with CLUTRR.

- CLUTRR is built on a static set of relations (22 family relations) while GraphLog can contain any number of such relations since it's a flexible generator along with a dataset.

**Synthetic Graph Generation**. Synthetic graph generation is extensively studied using various neural generative approaches for scale-free random graphs (You et al., 2018; Liao et al., 2019). Unlike these approaches, GraphLog is built using a procedural graph generator, which allows us to control the logical validity of the query graphs, which is defined in Section 3. Control over logical validity is easier in synthetically generated datasets, which are used extensively in Inductive Logic Programming (ILP) / Rule Learning context (Cornelio and Thost, 2019) [1]. GraphLog on the other hand, is targeted towards Graph Neural Network evaluation, which also supports for richer distribution shift evaluation through Multi-task learning and Continual learning scenarios.

## 3 GRAPHLOG

**Background and Terminology**. We leverage terminology and definitions from both knowledge graph and logic programming literature. A *graph* $G = (V_G, E_G)$ is a collection of a set of nodes $V_G$ and a set of edges $E_G$ between the nodes. We assume that the graphs are *relational*, meaning that each edge between two nodes (say $u$ and $v$) is assigned a *label*, and can be denoted as $r_i(u, v) \in E_G$. A *relation set* $\mathfrak{R}$ is a set of relations $\{r_1, r_2, ... r_K\}$.

---

[1] http://ilp16.doc.ic.ac.uk/competition

In logic programming terms, nodes in the graph correspond to *constants*, and edges correspond to *ground atoms*. Thus, as a slight abuse and mixture of notations, existence of an edge in the graph, i.e., $r_i(u, v) \in E_G$, is equivalent to background assumption that ground atom is true (i.e., that $r_i(u, v) \leftarrow$). We define a *rule set* $\mathcal{R}$ as set of *dyadic definite Datalog clauses* (Abiteboul et al., 1995) of form:

$$r_k(U, V) \leftarrow r_i(U, Z) \wedge r_j(Z, V). \tag{1}$$

Note that Equation 1 is also termed as *chain-rule* in logic literature. Following standard convention, we use upper-case to denote variables (e.g., the variables in Equation 1 can be substituted for nodes) and lower-case for constants (e.g., to denote nodes $v \in V_G$). The relations $r_i, r_j$ form the *body* while the relation $r_k$ forms the *head* of the rule. Path-based Horn clauses of this form represent a limited and well-defined subset of first-order logic. They encompass the types of logical rules learned by state-of-the-art rule induction systems for knowledge graph completion (Das et al., 2017; Evans and Grefenstette, 2017; Meilicke et al., 2018; Sadeghian et al., 2019; Teru et al., 2020; Yang et al., 2017; Zhang et al., 2019) and are thus a useful synthetic test-bed.

We use $p_G^{u,v}$ to denote a *path* from node $u$ to $v$ in a graph $G$ (i.e., a sequence unique nodes connected by edges). In logical terms, $p_G^{u,v}$ corresponds to the conjunction of all the edge predicates in the path. We construct graphs according to rules of the form in Equation 1 so that a path between two nodes will always imply a *specific* relation between these two nodes. In other words, if we let $\mathcal{P}_G$ denote the set of all finite paths in the graph $G$, then we have that

$$\forall p_G^{u,v} \in \mathcal{P}_G \, \exists r_i \in \mathfrak{R} : (p_G^{u,v} \rightarrow r_i(u, v)) \wedge (\forall r_j \in \mathfrak{R} \setminus \{r_i\} \, \neg(p_G^{u,v} \wedge r_j(u, v))). \tag{2}$$

Thus, by following the path between two nodes, and applying rules of the form of Equation 1 according to the edges of the path, we can always *resolve* the relationship between the nodes.

**Problem Setup**. Given this setup, the task is to predict the relation between a pair of nodes. In particular, we define a query $(g, u, v)$ as follows: (i) the *query subgraph* $g \subset G$ is a subgraph of the full graph $G$; (ii) the *query nodes* are present in the query subgraph (i.e., $u, v \in V_g$); (iii) the edge between the query nodes is *not* present in the query subgraph, but the edge *is* present in the underlying graph (i.e., $r_?(u, v) \in E_G, r_?(u, v) \notin E_g$). The goal is to infer the missing edge between query nodes $r_?(u, v)$. We assume that the learning agent has access to a set of training queries $g_1, ..., g_n \subset G$ to optimize a prediction model before being tested on a set of held-out queries $g_{n+1}, ..., g_{n+n'} \subset G$.

In knowledge graph terms, this is a form of link or relation prediction, as our goal is to infer the missing edge between two nodes. Unlike most work on knowledge graph completion, we emphasize an *inductive* problem setup, where the query graph $g$ in each training and testing example is unique, requiring generalization to unseen graphs (Teru et al., 2020).

Finally, note that we will sometimes refer to the full graphs $G$ as *world graphs*, in order to distinguish them from the *query graphs* $g \subset G$. We make this distinction because GraphLog involves link prediction over several different *logical worlds*, denoted $W = (G_W, \mathcal{R}_W)$, with each defined with by its own underlying world graph $G_W$ and rule set $\mathcal{R}_W$. Our setup enables controlling the overlap between the rules in these different logical worlds, allowing for a unique ability to quantify the logical generalization capabilities of a learning agent.

### 3.1 DATASET GENERATION

As discussed in Section 2, we want our proposed benchmark to provide four key desiderata: (i) interpretable rules, (ii) diversity, (iii) compositional generalization and (iv) large number of tasks. We describe how our dataset generation process ensures all four aspects.

**Rule generation**. We create a set $\mathfrak{R}$ of possible relations and use it to create a randomly generated rule set $\mathcal{R}$, such two key constraints are satisfied: **(i)** No two rules in $\mathcal{R}$ can have the same body, ensuring that Equation 2 is satisfied. **(ii)** Rules cannot have common relations among the *head* and *body*, ensuring the absence of cyclic dependencies in rules. Generating the dataset using a consistent and well-defined rule set ensures interpretability in the resulting dataset. The full algorithm for rule generation is given in Appendix (Algorithm 1).

**Graph generation**. The graph generation process has two steps: In the first step, we sample overlapping subsets of $\mathcal{R}_W \subset \mathcal{R}$ to create individual *worlds* (as shown in Figure 1). In each world, we recursively sample and use rules in $\mathcal{R}_W$ to generate the world graph $G_W$ for that world. This

sampling procedure creates a diverse set of world graphs by considering only certain subsets of $\mathcal{R}$. By controlling the extent of overlap between the subsets of $\mathcal{R}$ (in terms of the number of common rules), we can precisely control the similarity between the different worlds. The full algorithm for generating the world graphs and controlling the similarity between the *worlds* is given in Appendix (Algorithm 3 and Section A.2).

In the second step, the world graph $G_W$ in each world is used to sample a set of query graphs $G_W^S = (g_1, \cdots g_N)$ for that specific world (shown as Step (a) in Figure 2). A query graph $g_i$ is sampled from $G_W$ by sampling a pair of nodes $(u, v)$ from $G_W$ and then by sampling a path $p_{G_W}^{u,v}$. The edge $r_i(u, v)$ between the source and sink node of the path provides the target relation for the learning model to predict. To increase the complexity of the sampled $g_i$ graphs (beyond being simple paths), we also add nodes to $g_i$ by sampling neighbors of the nodes on $p_{G_W}^{u,v}$, such that no other shorter path exists between $u$ and $v$. Algorithm 4 (in the Appendix) details our graph sampling approach. Note that this sampling process leads to rich and complex graph structures, including cycles and motifs (see Figure 7(a) in Appendix for example of graphs). [2]

**GraphLog Dataset Suite**. We use the above data generation process to instantiate a dataset suite with 57 distinct logical *worlds* with 5000 query graphs per *world* (Figure 1). The dataset is split into train, validation, and test *worlds*. The query graphs within each *world* are also split into train, validation, and test sets (Table 2). Though we instantiate 57 *worlds*, the GraphLog generator can instantiate an arbitrary number of *worlds* and has been included in the supplementary material.

## 3.2 SETUPS SUPPORTED IN GRAPHLOG

**Supervised learning**. A model is trained (and evaluated) on the train (and test split) of a *world*. The number of rules grows exponentially with the number of relations $K$, making it impossible to train on all possible combinations of the relations. We expect that a *perfectly* systematic model inductively generalizes to unseen combinations of relations by training only on a subset of combinations.

**Multi-task learning**. GraphLog provides multiple logical *worlds*, with their training and evaluation splits. A model is trained on (and evaluated on) the train (and test) splits of several *worlds* $(W_1, \cdots, W_M)$. GraphLog enables us to control the complexity of each world and similarity between the worlds to evaluate how model performance varies across similar vs. dissimilar *worlds*. GraphLog is designed to study the effect of pre-training on adaptation. In this setup, the model is pre-trained on the train split of multiple *worlds* $(W_1, \cdots, W_M)$ and fine-tuned on the train split of the unseen heldout *worlds* $(W_{M+1}, \cdots, W_N)$. The model is evaluated on the *test* split of the heldout *worlds*. GraphLog enables mimicking in-distribution and out-of-distribution train (and test) scenarios and quantify the effect of multi-task pre-training for adaptation performance.

**Continual learning**. The model is trained on a sequence of *worlds*. Before training on a new *world*, it is evaluated on all the *worlds* that it has trained on so far. Given the challenges involved in continual learning (Thrun and Pratt, 2012; Parisi et al., 2019; De Lange et al., 2019; Sodhani et al., 2019), we do not expect the models to perform well on the previous tasks. Nonetheless, given that we are evaluating the models for relational reasoning and our datasets share *relations*, we expect the models to retain some knowledge of previous tasks. We use the performance on the previous tasks to infer if the models learn to solve the relational reasoning tasks or just *fit* to the current dataset distribution.

**Controlling similarity**. An unique feature of GraphLog is it allows practitioners to control the distribution shift when evaluating multi-task and continual learning baselines. In GraphLog the diversity of worlds can be controlled via similarity. Concretely, the similarity between two worlds $W^i$ and $W^j$ is defined as $\text{Sim}(W^i, W^j) = |\mathcal{R}^i \cap \mathcal{R}^j|$, where $W_i$ and $W_j$ are the graph worlds and $\mathcal{R}^i$ and $\mathcal{R}^j$ are the set of rules associated with them. Thus GraphLog enables various training scenarios - training on highly similar worlds or training on a mix of similar and dissimilar worlds. This fine grained control allows GraphLog to mimic both in-distribution and out-of-distribution scenarios - during training and testing. It also enables us to precisely categorize the effect of multi-task pre-training when the model needs to adapt to novel worlds.

---

[2]Due to the current formulation (Equation 2), GraphLog is restricted to have at most a single edge between a pair of nodes. This formulation is chosen to make the task simpler in order to analyze the effects of generalization in detail. In future work, we will be exploring the effect of multiple edges using this benchmark, which is a characteristic of many real-world knowledge graphs.

## 4 MODELS

In this section, we describe the message passing graph neural networks that we evaluate on GraphLog benchmark. These models take as input the query $(g_i, u, v)$ and a learnable relation embedding $\mathbf{r}_i \in \mathbb{R}^d$ (Step (d) and (e) in Figure 2), and output the relation as a k-way softmax (Step (f) in Figure 2). We use both attention-based and non-attention based models.

**Relational Graph Convolutional Network (RGCN)**. For a relational graph, the `RGCN` model (Schlichtkrull et al., 2018) is a natural choice for a baseline architecture. In this approach, we iterate a series of message passing operations: $\mathbf{h}_u^{(t)} = \text{ReLU}\left(\sum_{r_i \in R} \sum_{v \in \mathcal{N}_{r_i}(u)} \mathbf{r}_i \times_1 \mathcal{T} \times_3 \mathbf{h}_v^{(t-1)}\right)$, where $\mathbf{h}_u^{(t)} \in \mathbb{R}^d$ denotes representation of a node $u$ at the $t^{th}$ layer of the model, $\mathcal{T} \in \mathbb{R}^{d_r \times d \times d}$ is a learnable tensor, $\mathbf{r} \in \mathbb{R}^d$ is relation representation, and $\mathcal{N}_{r_i}(u)$ denotes the neighbors of $u$ (related by $r_i$). $\times_j, j = 1, 2, 3$ denotes multiplication across the modes of the tensor. `RGCN` model learns a relation-specific propagation matrix, specified by the interaction between relation embedding $\mathbf{r}_i$ and shared tensor $\mathcal{T}$.

**Edge-based Graph Attention Network (Edge-GAT)**. Many recent works have highlighted the importance of the attention mechanism, especially in the context of relational reasoning (Vaswani et al., 2017; Santoro et al., 2018; Schlag et al., 2019). Motivated by this observation, we investigate an extended version of the GAT (Veličković et al., 2017), in order to handle the edge types. We complement GAT by incorporating gating via an LSTM (Hochreiter and Schmidhuber, 1997) and where the attention is conditioned on both the incoming message (from the other nodes) and the relation embedding (of the other nodes): $\mathbf{m}_{\mathcal{N}(u)} = \sum_{r_i \in R} \sum_{v \in \mathcal{N}_{r_i}(u)} \alpha\left(\mathbf{h}_u^{(t-1)}, \mathbf{h}_v^{(t-1)}, \mathbf{r}\right), \mathbf{h}_u^{(t)} = \text{LSTM}(\mathbf{m}_{\mathcal{N}(u)}, \mathbf{h}_u^{(t-1)})$. Following the original `GAT` model, the attention function $\alpha$ uses concatenated input vectors. We refer to this model as the Edge GAT (`E-GAT`) model.

**Query and node representations**. We predict the relation for query $(g_i, u, v)$ by concatenating $\mathbf{h}_u^{(K)}, \mathbf{h}_v^{(K)}$ (final-layer query node embeddings) and applying a feedforward network (Step (f) in Figure 2). In absence of node features, we randomly initialize all the node embeddings in the GNNs.

**Representation functions** GNN models described above expect to learn $d$-dimensional relation representations. We can further re-structure them notation-wise into two modules: **a)** *Representation module* which is represented as a function $f_r : G_W \times R \to \mathbb{R}^d$, that maps logical relations within a particular *world* $W$ to $d$-dimensional vector representations, and **b)** *Composition module*, which is a function $f_c : G_W \times V_{G_W} \times V_{G_W} \times \mathbb{R}^{d \times |R|} \to R$, that learns to compose the relation representations learned by $f_r$ and answer queries over a knowledge graph. In this formulation, the relation embedding matrix $\mathbf{r}_i \in \mathbb{R}^d$ used in RGCN and Edge-GAT can be thought of the output of a *direct parameterization* module (`Param`), which is devoid of any input and simply returns the embeddings from a lookup table. Thus these two baselines can be redefined as `Param-RGCN` and `Param-E-GAT` models, where RGCN and E-GAT are the composition modules. A major limitation of this approach is that the relation representations are optimized specifically for each logical world, and there is no inductive bias towards learning representations that can generalize.

**Learning representations from the graph structure.** To define a more powerful representation function, we consider learning relation representations as a function of the graph $G_W$ underlying a logical world $W$. We consider an "extended" form of $G_W$,

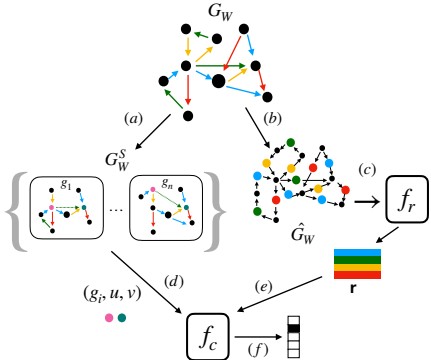

Figure 2: Overview of training process: **(a)**: Sample multiple graphs from $G_W$. **(b)**: Convert the relational graph into extended graph $\hat{G}_W$. Note that edges of different color (denoting different types of relations) are replaced by a node of same type in $\hat{G}_W$. **(c)**: Learn representations of the relations ($\mathbf{r}$) using $f_r$ (extended graph as the input). In case of `Param` models, the relation representations are parameterized via an embedding layer and the extended graph is not created. **(d, e)**: The composition function takes as input the query $g_i, u, v$ and the relational representation $\mathbf{r}$. **(f)**: The composition function predicts the relation between the nodes $u$ and $v$.

| $f_r$ | $f_c$ | S Accuracy | D Accuracy |
|-------|-------|------------|------------|
| GAT | E-GAT | **0.534** ±0.11 | **0.534** ±0.09 |
| GAT | RGCN | 0.474 ±0.11 | 0.502 ±0.09 |
| GCN | E-GAT | 0.522 ±0.1 | 0.533 ±0.09 |
| GCN | RGCN | 0.448 ±0.09 | 0.476 ±0.09 |
| Param | E-GAT | 0.507 ±0.09 | 0.5 ±0.09 |
| Param | RGCN | 0.416 ±0.07 | 0.449 ±0.07 |

Table 3: Multitask evaluation performance when trained on different data distributions (categorized based on their similarity of rules: Similar (S) containing similar worlds and a mix of similar and dissimilar worlds (D))

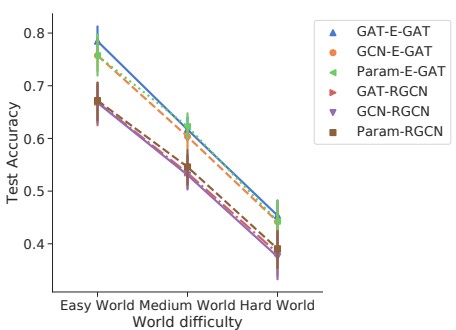

Figure 3: We categorize the datasets in terms of their relative *difficulty* (see Appendix). We observe that the models using E-GAT as the composition function consistently work well.

denoted $\hat{G}_W$, which introduces new nodes (called *edge-nodes*) corresponding to each edge in the original world graph $G_W$. For an edge $r(u, v) \in E_G$, the corresponding edge-node $(u - r - v)$ is connected to only those nodes that were incident to it in the original graph (i.e. nodes $u$ and $v$; see Figure 2, Step (b)). This new graph $\hat{G}_W$ only has one type of edge and comprises of nodes from both the original graph and from the set of edge-nodes. We learn the relation representations by training a GNN model on $\hat{G}_W$ and by averaging the edge-node embeddings corresponding to each relation type $r_i \in \mathfrak{R}$. (Step (c) in Figure 2). For the GNN model, we consider the Graph Convolutional Network (GCN) (Kipf and Welling, 2016) and the Graph Attention Network (GAT) architectures. The intuition behind creating the extended-graph is that the representation GNN function can learn the relation embeddings based on the structure of the complete relational graph $G_W$. Both the representation module and the composition modules are trained end-to-end to predict the relations. Thus, we end up with four more models to evaluate GraphLog , namely GCN-RGCN, GCN-E-GAT, GAT-RGCN and GAT-E-GAT.

## 5 EXPERIMENTS

We aim to quantify the performance of the different GNN models on the task of logical relation reasoning, in three setups: **(i)** Single Task Learning, **(ii)** Multi-Task Training and **(iii)** Continual Learning. Our experiments use the GraphLog benchmark with distinct 57 *worlds* (Section 3) and 6 different GNN models (Section 4). In the main paper, we highlight the key results and provide the full results in the Appendix. The code is included with the supplemental material.

### 5.1 SINGLE TASK SUPERVISED LEARNING

We evaluate the models on all the 57 *worlds*, one model-*world* pair at a time (342 combinations). With GraphLog , it is difficult for one model to outperform others on all the 57 datasets just by exploiting dataset-specific bias, thereby making the conclusions more robust. We categorize the *worlds* in three categories of *difficulty*—*easy*, *moderate* and *difficult*—based on the relative test performance of the models on each *world* and present the results in Figure 3. (A complimentary formulation of difficulty is explained in Appendix A.4). We observe that the models using E-GAT composition function always outperform the models using RGCN. This confirms our hypothesis about leveraging attention to improve the performance on relational reasoning tasks. Interestingly, the relative ordering among the *worlds*, in terms of the test accuracy of the different models, is consistent irrespective of the model we use, highlighting the intrinsic ordering of difficulty of the different *worlds* in GraphLog.

### 5.2 MULTI-TASK TRAINING

**Standard multi-task training**. First, we evaluate how changing similarity among training *worlds* affects test performance in the multi-task setup. We train a model on eight *worlds* and test on

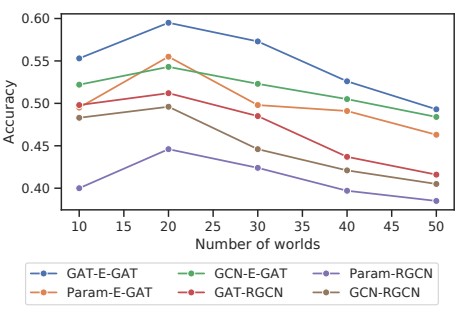

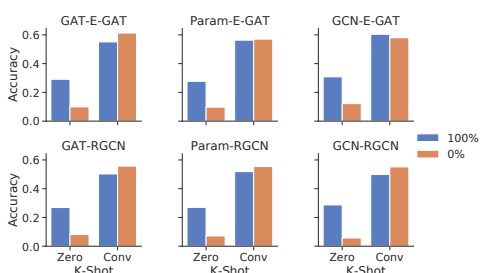

Figure 4: We run multitask experiments over an increasing number of worlds to stress the capacity of the models. We report the average of test set performance across the worlds that the model has trained on so far. All the models reach their optimal performance at 20 worlds, beyond which their performance starts to degrade.

Figure 5: We evaluate the effect of changing the similarity between the training and the evaluation datasets. The colors of the bars depicts how similar the two distributions are while the *y-axis* shows the mean accuracy of the model on the test split of the evaluation *world*. We report both the zero-shot adaptation performance and performance after convergence.

three distinct *worlds*. In Table 3, we observe that mix of similar and dissimilar *worlds* improves generalization capabilities of all the models. Similar to the trends observed in the single task setup, the `GAT-EGAT` model consistently performs either as good as or better than other models. The models using `EGAT` for the composition function perform better than the ones using the `RGCN` model. Figure 4 shows how the models' performance changes when we perform multi-task training on an increasingly large set of *worlds*. Interestingly, we see that model performance improves as the number of *worlds* is increased from 10 to 20 but then begins to decline, indicating model capacity saturation in the presence of too many diverse *worlds*.

**Multi-task pre-training**. In this setup, we pre-train the model on multiple *worlds* and adapt on a heldout *world*. We study how the models' adaption capabilities vary as the similarity between the training and the evaluation distributions changes. Figure 5 considers the case of zero-shot adaptation and adaptation till convergence. As we move along the *x-axis*, the zero-shot performance (shown with solid colors) decreases in all the setups. This is expected as the similarity between the training and the evaluation distributions also decreases. An interesting trend is that the model's performance, after adaptation, increases as the similarity between the two distributions decreases. This suggests that training over a diverse set of distributions improves adaptation capability.

### 5.3 Continual Learning Setup

We evaluate knowledge *retention* capability by training GNNs on a sequence of *worlds* (arranged by relative similarity). After convergence on any *world*, we report average of model's performance on all previous *worlds*. Figure 6(a) shows the rapid degradation in model's performance on the previous worlds as it trains on new *worlds*, highlighting the limitation of current models for continual learning.

**The role of the representation function**. We also investigate the model's performance in a continual learning setup where the model learns only a *world*-specific representation function or a *world*-specific composition function, and where the other module is shared across the worlds. In Figure 6(b), we observe that sharing the representation function reduces the effect of catastrophic forgetting, but sharing the composition function does not have the same effect. This suggests that the representation function learns representations that are useful across the *worlds*.

## 6 Discussion & Conclusion

In this work, we propose GraphLog, a benchmark suite for evaluating the logical generalization capabilities of GNNs. GraphLog is grounded in first-order logic and provides access to a large number of diverse tasks that require compositional generalization to solve, including single task supervised learning, multi-task learning, and continual learning. Our results highlight the importance of attention

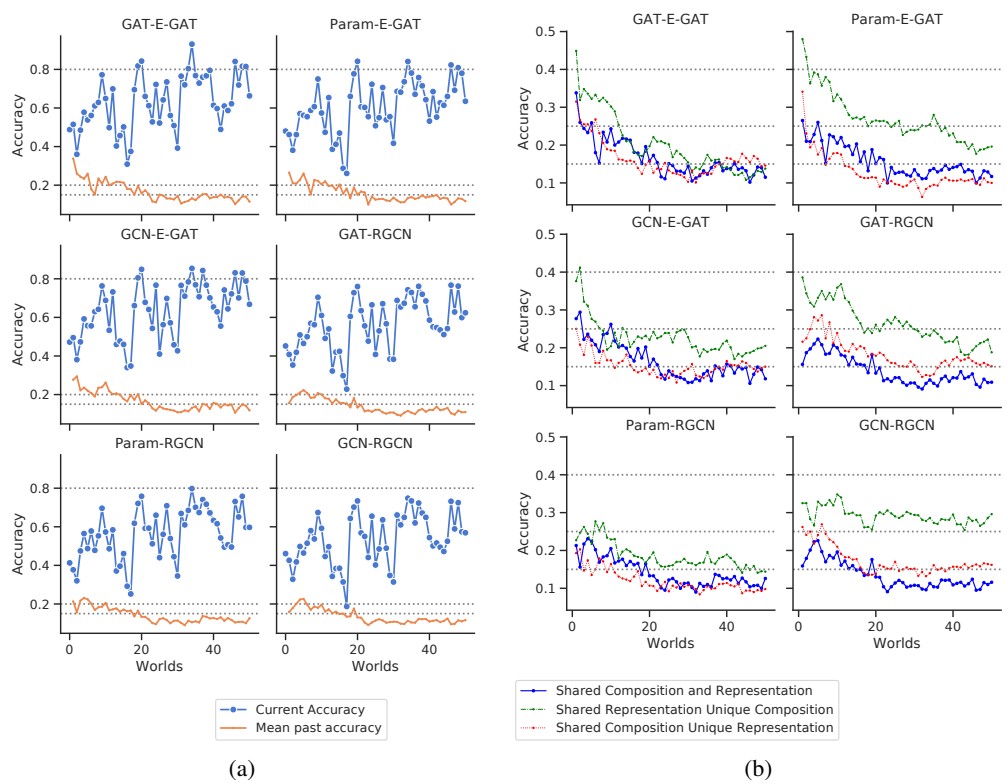

Figure 6: Evaluation of models in continual learning setup. In Figure 6(a), the blue curve shows the accuracy on the current *world* and the orange curve shows the mean of accuracy on all *previously* seen *worlds*. Training on new *worlds*, degrades model's performance on the past *worlds* (catastrophic forgetting). In Figure 6(b), sharing representation function reduces catastrophic forgetting.

mechanisms and modularity to achieve logical generalization, while also highlighting open challenges related to multi-task and continual learning in the context of GNNs. A natural direction for future work is leveraging GraphLog for studying *fast adaptation* and *meta-learning* in the context of logical reasoning (e.g., via gradient-based meta-learning), as well as integrating state-of-the-art methods (e.g., regularization techniques) to combat catastrophic forgetting in the context of GNNs.

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

# A  GRAPHLOG

## A.1  EXTENDED TERMINOLOGY

In this section, we extend the terminology introduced in Section 3 to introduce the term *descriptor*. A descriptor of a query graph $g$ is computed by concatenating the relations occurring on the shortest path from the source and sink in the query $(u, v)$. For example, if a query graph is described in the form of a list of edges such as $g \rightarrow (0, 1, r_1), (1, 2, r_2), (1, 3, r_3), (2, 4, r_4)$, where $r_i$ are the relations, then the descriptor of this graph given the query $(0, 4)$ would be $D_g = r_1, r_2, r_4$. In all our train, validation and test splits, we split graphs based on the unique set of descriptors, such as to maintain inductive reasoning aspect of the task.

## A.2  DATASET GENERATION

This section follows up on the discussion in Section 3.1. We describe all the steps involved in the dataset generation process.

**Rule Generation**. In Algorithm 1, we describe the complete process of generating rules in GraphLog . We require the set of $K$ relations, which we use to sample the rule set $\mathcal{R}$. Then, we iterate through all possible combinations of relations in DataLog format to sample possible candidate rules. We impose two constraints on the candidate rule: **(i)** No two rules in $\mathcal{R}$ can have the same body. This ensures consistency between the rules. **(ii)** Candidate rules cannot have common relations among the *head* and *body*. This ensures absence of cycles. We also add the inverse rule of our sampled candidate rule and check the same consistencies again. We employ two types of unary Horn clauses to perform the closure of the available rules and to check the consistency of the different rules in $\mathcal{R}$. Using this process, we ensure that all generated rules are sound and consistent with respect to $\mathcal{R}$.

**World Sampling**. From the set of rules in $\mathcal{R}$, we partition rules into buckets for different worlds (Algorithm 2). We use a simple policy of bucketing via a sliding window of width $w$ with stride $s$, to classify rules pertaining to each world. For example, two such consecutive worlds can be generated as $\mathcal{R}^t = [\mathcal{R}_i \ldots \mathcal{R}_{i+w}]$ and $\mathcal{R}^{t+1} = [\mathcal{R}_{i+s} \ldots \mathcal{R}_{i+w+s}]$. (Algorithm 2) We randomly permute $\mathcal{R}$ before bucketing in-order.

**Graph Generation**. This is a two-step process where first we sample a *world graph* (Algorithm 3) and then we sample individual graphs from the *world graph* (Algorithm 4). Given a set of rules $\mathcal{R}_\mathcal{S}$, in the first step, we recursively sample and apply rules in $\mathcal{R}_\mathcal{S}$ to generate a *relation graph* called *world graph*. This sampling procedure enables us to create a diverse set of *world graphs* by considering only certain subsets (of $\mathcal{R}$) during sampling. By controlling the extent of overlap between the subsets of $\mathcal{R}$ (in terms of the number of rules that are common across the subsets), we can precisely control the *similarity* between the different *world graphs*. By selecting subsets which have higher dissimilarity between each other, we introduce more diversity in terms of logical rules.

In the second step (Algorithm 4), the *world graph* is used to sample a set of query graphs $G_W^S = \{g_1, \cdots g_N\}$. A query graph $g_i$ is sampled from $G_W$ by sampling a pair of nodes $(u, v)$ from $G_W$ and then by sampling a *resolution path* $p_{G_W}^{u,v}$. The edge $r_i(u, v)$ provides the target relation that the learning model has to predict. Since the *relation* for the edge $r_i(u, v)$ can be *resolved* by composing the relations along the *resolution* path, the relation prediction task tests for the compositional generalization abilities of the models. We first sample all possible resolution paths and get their individual descriptors $D_{g_i}$, which we split in training, validation and test splits. We then construct the training, validation and testing graphs by first adding all edges of an individual $D_{g_i}$ to the corresponding query graph $g_i$, and then sampling neighbors of $p_{g_i}$. Concretely, we use Breadth First Search (BFS) to sample the neighboring subgraph of each node $u \in p_{g_i}$ with a decaying selection probability $\gamma$. This allows us to create diverse input graphs while having precise control over its resolution by its descriptor $D_{g_i}$. Splitting the dataset over these descriptor paths ensures inductive generalization.

## A.3  COMPUTING SIMILARITY

GraphLog provides precise control for categorizing the similarity between different worlds by computing the overlap of the underlying rules. Concretely, the similarity between two worlds $W^i$ and

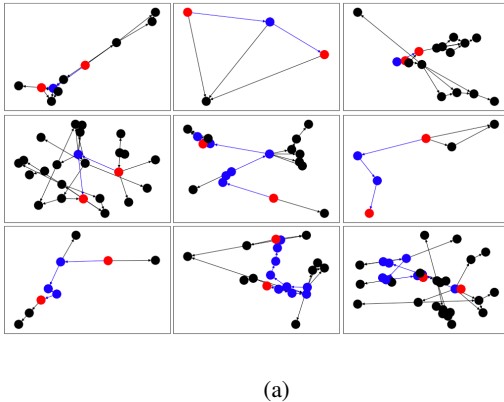 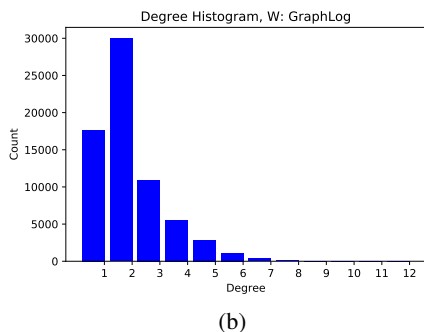

(a) (b)

Figure 7: Figure 7(a) represents graphs drawn at random from different worlds available in GraphLog . Nodes in red denote the query nodes, and the nodes in blue denote the shortest path among the query nodes. Figure 7(b) represents the degree distribution of all graphs from all worlds in GraphLog .

---

**Algorithm 1** Rule Generator

---

    **Input:** Set of $K$ relations $\{r_i\}_K, K > 0$
    Define an empty rule set $\mathcal{R}$
    **for all** $r_i \in \{r_i\}_K$ **do**
        **for all** $r_j \in \{r_i\}_K$ **do**
            **for all** $r_k \in \{r_i\}_K$ **do**
                Define candidate rule $t : [r_i, r_j] \implies r_k$
                **if** Cyclical rule, i.e. $r_i == r_k$ OR $r_j == r_k$ **then**
                    Reject rule
                **end if**
                **if** $t[body] \notin \mathcal{R}$ **then**
                    Add $t$ to $\mathcal{R}$
                **end if**
            **end for**
        **end for**
    **end for**
    Check and remove any further cyclical rules.

---

$W^j$ is defined as $\text{Sim}(W^i, W^j) = |\mathcal{R}^i \cap \mathcal{R}^j|$, where $W_i$ and $W_j$ are the graph worlds and $\mathcal{R}^i$ and $\mathcal{R}^j$ are the set of rules associated with them. Thus GraphLog enables various training scenarios - training on highly similar worlds or training on a mix of similar and dissimilar worlds. This fine grained control allows GraphLog to mimic both in-distribution and out-of-distribution scenarios - during training and testing. It also enables us to precisely categorize the effect of multi-task pre-training when the model needs to adapt to novel worlds.

---

**Algorithm 2** Partition rules into overlapping sets

---

**Require:** Rule Set $\mathcal{R}_{\mathcal{S}}$
**Require:** Number of worlds $n_w > 0$
**Require:** Number of rules per world $w > 0$
**Require:** Overlapping increment stride $s > 0$
    **for** $i = 0; i < |\mathcal{R}_{\mathcal{S}}| - w;$ **do**
        $\mathcal{R}_i = \mathcal{R}_{\mathcal{S}}[i; i + w]$
        $i = i + s$
    **end for**

---

---

**Algorithm 3** World Graph Generator

---

**Require:** Set of relations $\{r_i\}_K, K > 0$
**Require:** Set of rules derived from $\{r_i\}_K, |\mathcal{R}| > 0$
**Require:** Set rule selection probability gamma $\gamma = 0.8$
   Set rule selection probability $P[\mathcal{R}[i]] = 1, \forall i \in |\mathcal{R}|$
**Require:** Maximum number of expansions $s \geq 2$
**Require:** Set of available nodes $N$, s.t. $|N| \geq 0$
**Require:** Number of cycles of generation $c \geq 0$
   Set *WorldGraph* set of edges $G_m = \emptyset$
   **while** $|N| > 0$ or $c > 0$ **do**
      Randomly choose an expansion number for this cycle: steps $= \text{rand}(2, s)$
      Set added edges for this cycle $E_c = \emptyset$
      **for all** step in steps **do**
         **if** step $= 0$ **then**
            With uniform probability, either:
            Sample $r_t$ from $\mathcal{R}_\mathcal{S}[head]$ and sample $u, v \in N$ without replacement, OR
            Sample an edge $(u, r_t, v)$ from $G_m$
            Add $(u, r_t, v)$ to $E_c$ and $G_m$
         **else**
            Sample an edge $(u, r_t, v)$ from $E_c$
         **end if**
         Sample a rule $\mathcal{R}[i]$ from $\mathcal{R}$ following $P$ s.t. $[r_i, r_j] \implies r_t$
         $P[\mathcal{R}[i]] = P[\mathcal{R}[i]] * \gamma$
         Sample a new node $y \in N$ without replacement
         Add edge $(u, r_i, y)$ to $E_c$ and $G_m$
         Add edge $(y, r_j, v)$ t $E_c$ and $G_m$
      **end for**
      **if** All rules in $\mathcal{R}$ is used atleast once **then**
         Increment $c$ by 1
         Reset rule selection probability $P[\mathcal{R}[i]] = 1, \forall i \in |\mathcal{R}|$
      **end if**
   **end while**

---

## A.4 COMPUTING DIFFICULTY

Recent research in multitask learning has shown evidence that models prioritize selection of difficult tasks over easy tasks while learning to boost the overall performance (Guo et al., 2018). Thus, GraphLog also provides a method to examine how pretraining on tasks of different difficulty level affects the adaptation performance. Due to the stochastic effect of partitioning of the rules, GraphLog consists of datasets with varying range of difficulty. We use the supervised learning scores (Table 6) as a proxy to determine the the relative difficulty of different datasets. We cluster the datasets such that tasks with prediction accuracy greater than or above 70% are labeled as *easy* difficulty, 50-70% are labeled as *medium* difficulty and below 50% are labeled as *hard* difficulty dataset. We find that the labels obtained by this criteria are consistent across the different models (Figure 3).

**Graph properties affecting difficulty**. While we compute difficulty based on the proxy of supervised learning scores, we observe that the relative difficulty of the tasks are highly correlated with the number of *descriptors* (Section A.1) available for each task. We can control the distribution of the dataset by explicitly requiring diversity in the descriptors. Worlds having less number of descriptors are prone to be more difficult for the task. This is due to the fact that will less available descriptors with respect to the budget of data samples, our generation module samples the same set of descriptors while adding variable noise. Thus, datasets with low descriptor count ends up with more relative noise. This shows that for a learner, a dataset with enough variety among the resolution paths of the graphs with less noise is relatively easier to learn compared to the datasets which has less variation and more noise.

---

**Algorithm 4** Graph Sampler

---

**Require:** Rule Set $\mathcal{R}_\mathcal{S}$
**Require:** World Graph $G_m = (V_m, E_m)$
**Require:** Maximum Expansion length $e > 2$
  Set Descriptor set $S = \emptyset$
  **for all** $u, v \in E_m$ **do**
    Get all walks $Y_{(u,v)} \in G_m$ such that $|Y_{(u,v)}| \leq e$
    Get all descriptors $D_{Y_{(u,v)}}$ for all walks $Y_{(u,v)}$
    Add $D_{Y_{(u,v)}}$ to $S$
  **end for**
  Set train graph set $G_{train} = \emptyset$
  Set test graph set $G_{test} = \emptyset$
  Split descriptors in train and test split, $S_{train}$ and $S_{test}$
  **for all** $D_i \in S_{train}$ or $S_{test}$ **do**
    Set source node $u_s = D_i[0]$ and sink node $v_s = D_i[-1]$
    Set prediction target $t = E_m[u_s][v_s]$
    Set graph edges $g_i = \emptyset$
    Add all edges from $D_i$ to $g_i$
    **for all** $u, v \in D_i$ **do**
      Sample Breadth First Search connected nodes from $u$ and $v$ with decaying probability $\gamma$
      Add the sampled edges to $g_i$
    **end for**
    Remove edges in $g_i$ which create shorter paths between $u_s$ and $v_s$
    Add $(g_i, u_s, v_s, t)$ to either $G_{train}$ or $G_{test}$
  **end for**

---

# B   SUPERVISED LEARNING ON GRAPHLOG

We perform extensive experiments over *all* the datasets available in GraphLog (statistics given in Table 6). We observe that in general, for the entire set of 57 worlds, the GAT_E-GAT model performs the best.

# C   MULTITASK LEARNING

## C.1   MULTITASK LEARNING ON DIFFERENT DATA SPLITS BY DIFFICULTY

|  |  | Easy | Medium | Difficult |
|---|---|---|---|---|
| $f_r$ | $f_c$ | Accuracy | Accuracy | Accuracy |
| GAT | E-GAT | **0.729** ±0.05 | **0.586** ±0.05 | 0.414 ±0.07 |
| Param | E-GAT | 0.728 ±0.05 | 0.574 ±0.06 | 0.379 ±0.06 |
| GCN | E-GAT | 0.713 ±0.05 | 0.55 ±0.06 | 0.396 ±0.05 |
| GAT | RGCN | 0.695 ±0.04 | 0.53 ±0.03 | **0.421** ±0.06 |
| Param | RGCN | 0.551 ±0.08 | 0.457 ±0.05 | 0.362 ±0.05 |
| GCN | RGCN | 0.673 ±0.05 | 0.514 ±0.04 | 0.396 ±0.06 |

Table 4: Inductive performance on data splits marked by difficulty

In Section A.4 we introduced the notion of *difficulty* among the tasks available in GraphLog . Here, we consider a set of experiments where we perform multitask training and inductive testing on the worlds bucketized by their relative difficulty (Table 4). We sample equal number of worlds from each difficulty bucket, and separately perform multitask training and testing. We evaluate the average prediction accuracy on the datasets within each bucket. We observe that the average multitask performance *also* mimics the relative task difficulty distribution. We find GAT-E-GAT model outperforms other baselines in *Easy* and *Medium* setup, but is outperformed by GAT-RGCN model in the *Difficult* setup. For each model, we used the same architecture and hyperparameter settings across the buckets. Optimizing individually for each bucket may improve the relative performance.

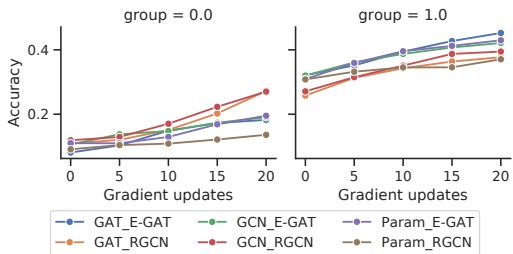

Figure 8: We perform fine-grained analysis of *few shot* adaptation capabilities in Multitask setting. Group 0.0 and 1.0 corresponds to 0% and 100% similarity respectively.

| $f_r$ | $f_c$ | Easy Accuracy | Medium Accuracy | Difficult Accuracy |
|---|---|---|---|---|
| GAT | E-GAT | $0.531 \pm 0.03$ | **0.569** $\pm 0.01$ | **0.555** $\pm 0.04$ |
| Param | E-GAT | $0.520 \pm 0.02$ | $0.548 \pm 0.01$ | $0.540 \pm 0.01$ |
| GCN | E-GAT | **0.555** $\pm 0.01$ | $0.561 \pm 0.02$ | $0.558 \pm 0.01$ |
| GAT | RGCN | $0.502 \pm 0.02$ | $0.532 \pm 0.01$ | $0.532 \pm 0.01$ |
| Param | RGCN | $0.535 \pm 0.01$ | $0.506 \pm 0.04$ | $0.539 \pm 0.04$ |
| GCN | RGCN | $0.481 \pm 0.02$ | $0.516 \pm 0.02$ | $0.520 \pm 0.01$ |
| Mean | | 0.521 | **0.540** | 0.539 |

Table 5: Convergence performance on 3 held out datasets when pre-trained on easy, medium and hard training datasets

## C.2 MULTITASK PRE-TRAINING BY TASK SIMILARITY

In the main paper (Section 5.2) we introduce the setup of performing multitask pre-training on GraphLog datasets and adaptation on the datasets based on relative similarity. Here, we perform fine-grained analysis of *few-shot* adapatation capabilities of the models. We analyze the adaptation performance in two settings - when the adaptation dataset has complete overlap of rules with the training datasets (*group*=1.0) and when the adaptation dataset has zero overlap with the training datasets (*group*=0.0). We find RGCN family of models with a graph based representation function has faster adaptation on the dissimilar dataset, with `GCN-RGCN` showing the fastest improvement. However on the similar dataset the models follow the ranking of the supervised learning experiments, with `GAT-EGAT` model adapting comparitively better.

## C.3 MULTITASK PRE-TRAINING BY TASK DIFFICULTY

Using the notion of *difficulty* introduced in Section A.4, we perform the suite of experiments to evaluate the effect of pre-training on *Easy*, *Medium* and *Difficult* datasets. Interestingly, we find the performance on convergence is better on Medium and Hard datasets on pre-training, compared to the Easy dataset (Table 5). This behaviour is also mirrored in k-shot adaptation performance (Figure 9), where pre-training on Hard dataset provides faster adaptation performance on 4/6 models.

## D CONTINUAL LEARNING

A natural question arises following our continual learning experiments in Section 5.3 : does the *order* of difficulty of the worlds matter? Thus, we perform an experiment following Curriculum Learning (Bengio et al., 2009) setup, where the order of the worlds being trained is determined by their relative difficulty (which is determined by the performance of models in supervised learning setup, Table 6, i.e., we order the worlds from easier worlds to harder worlds). We observe that while the current task accuracy follows the trend of the difficulty of the worlds (Figure 10(a)), the mean of past accuracy is significantly worse. This suggests that a curriculum learning strategy might not be optimal to learn graph representations in a continual learning setting. We also performed the same experiment with sharing only the composition and representation functions (Figure 10(b)), and observe similar trends where sharing the representation function reduces the effect of catastrophic forgetting.

| World ID | NC | ND | Split | ARL | AN | AE | D | M1 | M2 | M3 | M4 | M5 | M6 |
|---|---|---|---|---|---|---|---|---|---|---|---|---|---|
| rule_0 | 17 | 286 | train | 4.49 | 15.487 | 19.295 | Hard | 0.481 | 0.500 | 0.494 | 0.486 | 0.462 | 0.462 |
| rule_1 | 15 | 239 | train | 4.10 | 11.565 | 13.615 | Hard | 0.432 | 0.411 | 0.428 | 0.406 | 0.400 | 0.408 |
| rule_2 | 17 | 157 | train | 3.21 | 9.809 | 11.165 | Hard | 0.412 | 0.357 | 0.373 | 0.347 | 0.347 | 0.319 |
| rule_3 | 16 | 189 | train | 3.63 | 11.137 | 13.273 | Hard | 0.429 | 0.404 | 0.473 | 0.373 | 0.401 | 0.451 |
| rule_4 | 16 | 189 | train | 3.94 | 12.622 | 15.501 | Medium | 0.624 | 0.606 | 0.619 | 0.475 | 0.481 | 0.595 |
| rule_5 | 14 | 275 | train | 4.41 | 14.545 | 18.872 | Hard | 0.526 | 0.539 | 0.548 | 0.429 | 0.461 | 0.455 |
| rule_6 | 16 | 249 | train | 5.06 | 16.257 | 20.164 | Hard | 0.528 | 0.514 | 0.536 | 0.498 | 0.495 | 0.476 |
| rule_7 | 17 | 288 | train | 4.47 | 13.161 | 16.333 | Medium | 0.613 | 0.558 | 0.598 | 0.487 | 0.486 | 0.537 |
| rule_8 | 15 | 404 | train | 5.43 | 15.997 | 19.134 | Medium | 0.627 | 0.643 | 0.629 | 0.523 | 0.563 | 0.569 |
| rule_9 | 19 | 1011 | train | 7.22 | 24.151 | 32.668 | Easy | 0.758 | 0.744 | 0.739 | 0.683 | 0.651 | 0.623 |
| rule_10 | 18 | 524 | train | 5.87 | 18.011 | 22.202 | Medium | 0.656 | 0.654 | 0.663 | 0.596 | 0.563 | 0.605 |
| rule_11 | 17 | 194 | train | 4.29 | 11.459 | 13.037 | Medium | 0.552 | 0.525 | 0.533 | 0.445 | 0.456 | 0.419 |
| rule_12 | 15 | 306 | train | 4.14 | 11.238 | 12.919 | Easy | 0.771 | 0.726 | 0.603 | 0.511 | 0.561 | 0.523 |
| rule_13 | 16 | 149 | train | 3.58 | 11.238 | 13.549 | Hard | 0.453 | 0.402 | 0.419 | 0.347 | 0.298 | 0.344 |
| rule_14 | 16 | 224 | train | 4.14 | 11.371 | 13.403 | Hard | 0.448 | 0.457 | 0.401 | 0.314 | 0.318 | 0.332 |
| rule_15 | 14 | 224 | train | 3.82 | 12.661 | 15.105 | Hard | 0.494 | 0.423 | 0.501 | 0.402 | 0.397 | 0.435 |
| rule_16 | 16 | 205 | train | 3.59 | 11.345 | 13.293 | Hard | 0.318 | 0.332 | 0.292 | 0.328 | 0.306 | 0.291 |
| rule_17 | 17 | 147 | train | 3.16 | 8.163 | 8.894 | Hard | 0.347 | 0.308 | 0.274 | 0.164 | 0.161 | 0.181 |
| rule_18 | 18 | 923 | train | 6.63 | 25.035 | 33.080 | Easy | 0.700 | 0.680 | 0.713 | 0.650 | 0.641 | 0.618 |
| rule_19 | 16 | 416 | train | 6.10 | 17.180 | 20.818 | Easy | 0.790 | 0.774 | 0.777 | 0.731 | 0.729 | 0.702 |
| rule_20 | 20 | 2024 | train | 8.63 | 34.059 | 45.985 | Easy | 0.830 | 0.799 | 0.854 | 0.756 | 0.741 | 0.750 |
| rule_21 | 13 | 272 | train | 4.58 | 10.559 | 11.754 | Medium | 0.621 | 0.610 | 0.632 | 0.531 | 0.516 | 0.580 |
| rule_22 | 17 | 422 | train | 5.21 | 16.540 | 20.681 | Medium | 0.586 | 0.593 | 0.628 | 0.530 | 0.506 | 0.573 |
| rule_23 | 15 | 383 | train | 4.97 | 17.067 | 21.111 | Hard | 0.508 | 0.522 | 0.493 | 0.455 | 0.473 | 0.476 |
| rule_24 | 18 | 879 | train | 6.33 | 21.402 | 26.152 | Easy | 0.706 | 0.704 | 0.743 | 0.656 | 0.641 | 0.638 |
| rule_25 | 15 | 278 | train | 3.84 | 11.093 | 12.775 | Hard | 0.424 | 0.419 | 0.382 | 0.358 | 0.345 | 0.412 |
| rule_26 | 15 | 352 | train | 4.71 | 14.157 | 17.115 | Medium | 0.565 | 0.534 | 0.532 | 0.466 | 0.461 | 0.499 |
| rule_27 | 16 | 393 | train | 4.98 | 14.296 | 16.499 | Easy | 0.713 | 0.714 | 0.722 | 0.632 | 0.604 | 0.647 |
| rule_28 | 16 | 391 | train | 4.82 | 17.551 | 21.897 | Medium | 0.575 | 0.564 | 0.571 | 0.503 | 0.499 | 0.552 |
| rule_29 | 16 | 144 | train | 3.87 | 10.193 | 11.774 | Hard | 0.468 | 0.445 | 0.475 | 0.325 | 0.336 | 0.389 |
| rule_30 | 17 | 177 | train | 3.51 | 10.270 | 11.764 | Hard | 0.381 | 0.426 | 0.382 | 0.357 | 0.316 | 0.336 |
| rule_31 | 19 | 916 | train | 5.90 | 20.147 | 26.562 | Easy | 0.788 | 0.789 | 0.770 | 0.669 | 0.674 | 0.641 |
| rule_32 | 16 | 287 | train | 4.66 | 16.270 | 20.929 | Medium | 0.674 | 0.671 | 0.700 | 0.621 | 0.594 | 0.615 |
| rule_33 | 18 | 312 | train | 4.50 | 14.738 | 18.266 | Medium | 0.695 | 0.660 | 0.709 | 0.710 | 0.679 | 0.668 |
| rule_34 | 18 | 504 | train | 5.00 | 15.345 | 18.614 | Easy | 0.908 | 0.888 | 0.906 | 0.768 | 0.762 | 0.811 |
| rule_35 | 19 | 979 | train | 6.23 | 21.867 | 28.266 | Easy | 0.831 | 0.750 | 0.782 | 0.680 | 0.700 | 0.662 |
| rule_36 | 19 | 252 | train | 4.66 | 13.900 | 16.613 | Easy | 0.742 | 0.698 | 0.698 | 0.659 | 0.627 | 0.651 |
| rule_37 | 17 | 260 | train | 4.00 | 11.956 | 14.010 | Easy | 0.843 | 0.826 | 0.826 | 0.673 | 0.698 | 0.716 |
| rule_38 | 17 | 568 | train | 5.21 | 15.305 | 20.075 | Easy | 0.748 | 0.762 | 0.733 | 0.644 | 0.630 | 0.719 |
| rule_39 | 15 | 182 | train | 3.98 | 12.552 | 14.800 | Easy | 0.737 | 0.642 | 0.635 | 0.592 | 0.603 | 0.587 |
| rule_40 | 17 | 181 | train | 3.69 | 11.556 | 14.437 | Medium | 0.552 | 0.584 | 0.575 | 0.525 | 0.472 | 0.479 |
| rule_41 | 15 | 113 | train | 3.58 | 10.162 | 11.553 | Medium | 0.619 | 0.601 | 0.626 | 0.490 | 0.468 | 0.470 |
| rule_42 | 14 | 95 | train | 2.96 | 8.939 | 9.751 | Hard | 0.511 | 0.472 | 0.483 | 0.386 | 0.393 | 0.395 |
| rule_43 | 16 | 162 | train | 3.36 | 11.077 | 13.337 | Medium | 0.622 | 0.567 | 0.579 | 0.473 | 0.482 | 0.437 |
| rule_44 | 18 | 705 | train | 4.75 | 15.310 | 18.172 | Hard | 0.538 | 0.561 | 0.603 | 0.498 | 0.519 | 0.450 |
| rule_45 | 15 | 151 | train | 3.39 | 9.127 | 10.001 | Medium | 0.569 | 0.580 | 0.592 | 0.535 | 0.524 | 0.524 |
| rule_46 | 19 | 2704 | train | 7.94 | 31.458 | 43.489 | Easy | 0.850 | 0.820 | 0.828 | 0.773 | 0.762 | 0.749 |
| rule_47 | 18 | 647 | train | 6.66 | 22.139 | 27.789 | Easy | 0.723 | 0.667 | 0.708 | 0.620 | 0.649 | 0.611 |
| rule_48 | 16 | 978 | train | 6.15 | 17.802 | 21.674 | Easy | 0.812 | 0.798 | 0.812 | 0.772 | 0.763 | 0.753 |
| rule_49 | 14 | 169 | train | 3.41 | 9.983 | 11.177 | Easy | 0.714 | 0.734 | 0.700 | 0.511 | 0.491 | 0.615 |
| rule_50 | 16 | 286 | train | 3.99 | 12.274 | 16.117 | Medium | 0.651 | 0.653 | 0.656 | 0.555 | 0.583 | 0.570 |
| rule_51 | 16 | 332 | valid | 4.44 | 16.384 | 21.817 | Medium | 0.746 | 0.742 | 0.738 | 0.667 | 0.657 | 0.689 |
| rule_52 | 17 | 351 | valid | 4.81 | 16.231 | 20.613 | Medium | 0.697 | 0.716 | 0.754 | 0.653 | 0.655 | 0.670 |
| rule_53 | 15 | 165 | valid | 3.65 | 10.838 | 12.378 | Hard | 0.458 | 0.464 | 0.525 | 0.334 | 0.364 | 0.373 |
| rule_54 | 13 | 303 | test | 5.25 | 13.503 | 15.567 | Medium | 0.638 | 0.623 | 0.603 | 0.587 | 0.586 | 0.555 |
| rule_55 | 16 | 293 | test | 4.83 | 16.444 | 20.944 | Medium | 0.625 | 0.582 | 0.578 | 0.561 | 0.528 | 0.571 |
| rule_56 | 15 | 241 | test | 4.40 | 14.010 | 16.702 | Medium | 0.653 | 0.681 | 0.692 | 0.522 | 0.513 | 0.550 |
| AGG | 16.33 | 428.94 | | 4.70 | 14.89 | 18.37 | | **0.618 / 26** | 0.603 / 10 | 0.611 / 20 | 0.530 / 1 | 0.526 / 0 | 0.539 / 0 |

Table 6: Results on Single-task supervised setup for all datasets in GraphLog. Abbreviations: **NC**: Number of Classes, **ND**: Number of Descriptors, **ARL**: Average Resolution Length, **AN**: Average number of nodes, **AE**: Average number of edges
, **D**: Difficulty, **AGG**: Aggregate Statistics. List of models considered : **M1**: GAT-EGAT, **M2**: GCN-E-GAT, **M3**: Param-E-GAT, **M4**: GAT-RGCN, **M5**: GCN-RGCN and **M6**: Param-RGCN. Difficulty is calculated by taking the scores of the model (M1) and partitioning the worlds according to their accuracy ($\geq 0.7$ = Easy, $\geq 0.54$ and $< 0.7$ = Medium, and $< 0.54$ = Hard). We provide both the mean of the raw accuracy scores for all models, as well as the number of times the model is ranked first in all the tasks.

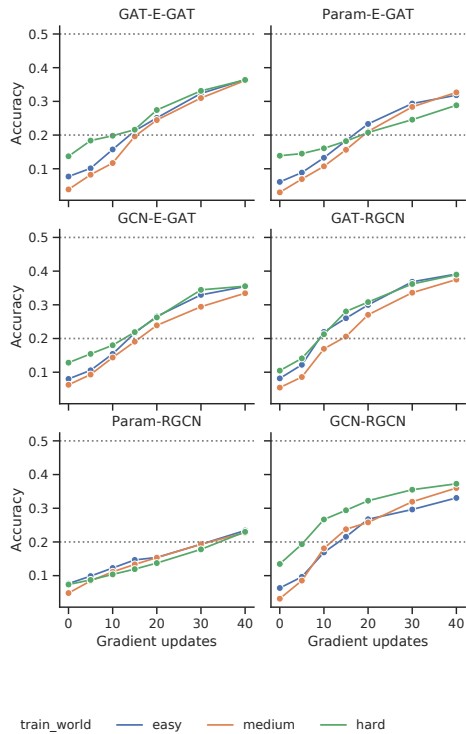

Figure 9: We evaluate the effect of $k$-shot adaptation on held out datasets when pre-trained on *easy*, *medium* and *hard* training datasets, among the different model architectures. Here, $k$ ranges from 0 to 40.

## E HYPERPARAMETERS AND EXPERIMENTAL SETUP

In this section, we provide detailed hyperparameter settings for both models and dataset generation for the purposes of reproducibility. The codebase and dataset used in the experiments are attached with the Supplementary materials, and will be made public on acceptance.

### E.1 DATASET HYPERPARAMS

We generate GraphLog with 20 relations or classes ($K$), which results in 76 rules in $\mathcal{R}_{\mathcal{S}}$ after consistency checks. For unary rules, we specify half of the relations to be symmetric and other half to have their invertible relations. To split the rules for individual worlds, we choose the number of rules for each world $w = 20$ and stride $s = 1$, and end up with 57 worlds $\mathcal{R}_0 \ldots \mathcal{R}_{56}$. For each world $\mathcal{R}_i$, we generate 5000 training, 1000 testing and 1000 validation graphs.

### E.2 MODEL HYPERPARAMS

For all models, we perform hyper-parameter sweep (grid search) to find the optimal values based on the validation accuracy. For all models, we use the relation embedding and node embedding to be 200 dimensions. Since the nodes in GraphLog does not have any features or attributes, we randomly initialize the embeddings in the GNN message passing layers for each epoch for all experiments. We train all models with Adam optimizer with learning rate 0.001 and weight decay of 0.0001. For supervised setting, we train all models for 500 epochs, and we add a scheduler for learning rate to decay it by 0.8 whenever the validation loss is stagnant for 10 epochs. In multitask setting, we sample a new task every epoch from the list of available tasks. Here, we run all models for 2000 epochs when we have the number of tasks $\leq 10$. For larger number of tasks (Figure 4), we train by proportionally increasing the number of epochs compared to the number of tasks. (2k epochs for 10 tasks, 4k epochs for 20 tasks, 6k epochs for 30 tasks, 8k epochs for 40 tasks and 10k epochs for

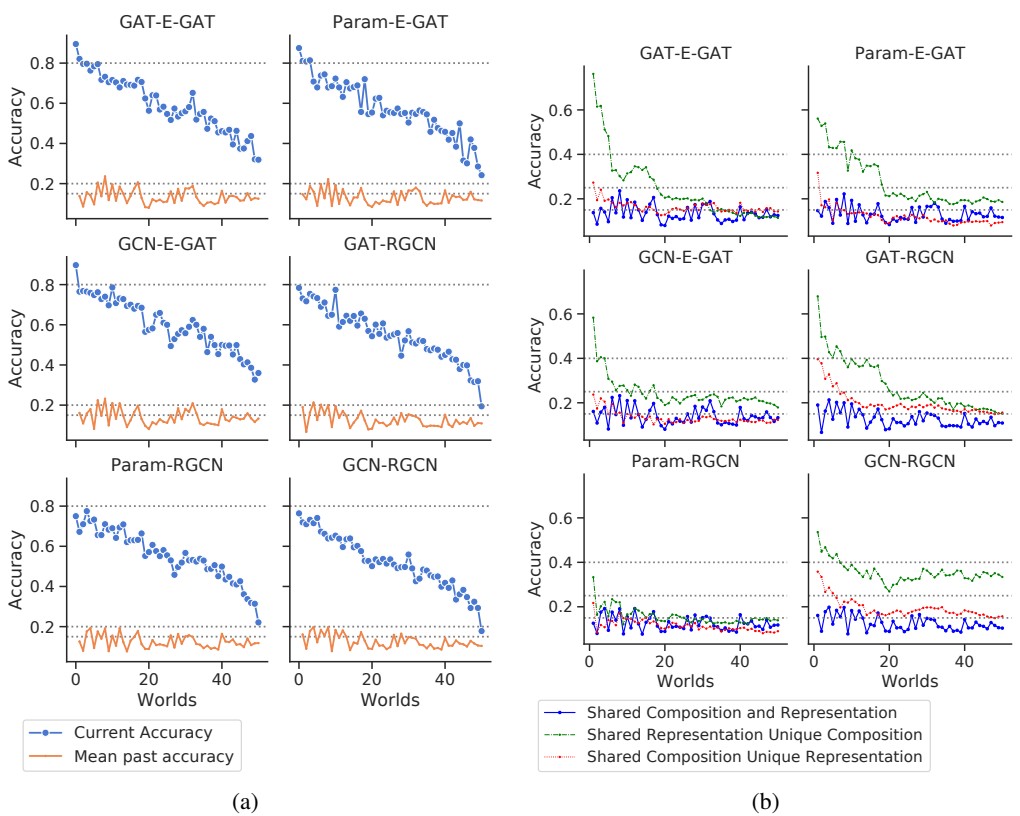

Figure 10: Curriculum Learning strategy in Continual Learning setup of GraphLog. Figure 10(a) presents the current task accuracy (blue) and mean of all previous task accuracy (orange). Figure 10(b) presents the mean of previous task accuracy when either the composition function or the representation function is shared for all worlds.

50 tasks). For continual learning experiment, we train each task for 100 epochs for all models. No learning rate scheduling is used for either multitask or continual learning experiments. Individual model hyper-parameters are as follows:

- Representation functions :
  - `GAT` : Number of layers = 2, Number of attention heads = 2, Dropout = 0.4
  - `GCN` : Number of layers = 2, with symmetric normalization and bias, no dropout
- Composition functions:
  - `E-GAT`: Number of layers = 6, Number of attention heads = 2, Dropout = 0.4
  - `RGCN`: Number of layers = 2, no dropout, with bias.

