# OpenReview forum: "GraphLog: A Benchmark for Measuring Logical Generalization in Graph Neural Networks"
_ICLR.cc/2021/Conference — Reject_

### Official Review · AnonReviewer4 · 2020-10-26
**Official Blind Review #4**

**Rating:** 5
**Confidence:** 4

**Review:**

The authors propose a benchmark for evaluating to which extend GNNs are able to reason in terms of logical/symbolic rules. The results show that standard GNNs are mostly lacking this capability (what is not really surprising). The paper is fairly understandable and the idea is interesting. The design of the benchmark makes sense to me. Since the writing could be more clear/thorough and I have issues with related work/systems that are not considered (details below), I however do not vote for acceptance.

(+) The benchmark consists of various datasets, interesting settings (e.g., multi-task training and continual learning), and proposes a small extension of GAT which improves performance.

(-) Logic Part:
- There's a mix of first-order logic (FOL) terminology (constants, relations) and Semantic Web terminology (entities, relations). I suggest to use either constant and predicate or entity and relation. In FOL, a relation (the notion the paper uses for predicates) is the interpretation of a predicate.
-  Rule (1) is in the most typical format considered in rule learning and called a "chain rule". You should mention this at least besides dyadic definite datalog, since the latter seems to be not fully covered. Also, other parts of the paper mention unary rules, which are not covered by (1).
- I do not think assumption (2) is realistic and wonder in how far it helps with the evaluation.

(-) Related Work: The paper mentions several related works, especially benchmarks. However, I do not understand why the authors completely ignore the entire field of rule learning/ILP. While this area is not solely based on deep learning approaches, the research and insights (esp. how to evaluate logical generalization) are relevant. There are even existing rule learning approaches based on deep learning. The paper compares to benchmarks considering image/text data but does not even mention rule-based benchmarks such as the ILP competition or Rudas (see below links). I would not support the paper's claim as it is written: "GraphLog is the only dataset specifically designed to test logical generalization capabilities on graph data, whereas previous works have largely focused on the image and text modalities." The similarities and differences to existing benchmarks (I agree that there are some, e.g., support for continual learning) need to be pointed out.
- http://ilp16.doc.ic.ac.uk/competition
- https://github.com/IBM/RuDaS

(-) Experiments: While it is ok to generally focus on GNNs, the evaluation should consider established rule-learning approaches to set the context (e.g., the recently published Open Graph Benchmark also considers standard baselines), in particular, given that the GNNs perform so disappointing. There are competitive rule learning systems such as AMIE and also deep-learning based systems which could serve as baselines.

----------------------------------------------
Smaller Comments:
- p.2 What is the difference between generalization and adaptation?
- I think it might be good to define the notion of "compositional generalization" more clearly.
- Table. 3 caption: I think the definition of rule similarity is important enough to come in the main paper.
- p.17 ?? Fig ref
- A.2 "we ensure that all generated rules are sound and consistent with respect to R." - why should a rule not be sound or consistent w.r.t R?

----------------------------------------------
Update after Rebuttal: I have read the other reviews and authors' responses but do no change my scores.

I still share the impression of Reviewer 3 that several choices in the design of the benchmark seem to be very restricted and arbitrary (e.g., the single possible derivation sequence).

Related work in terms of rule learning has been incorporated partly now, but there is no proper comparison -- especially in terms of the dimensions mentioned in the previous item. Also, there have been proposals similar to the E-GAT model the authors introduce which are completely ignored in the paper (e.g., MEMORY-BASED GRAPH NETWORKS, ICLR 2020; Graph Neural Networks for Social Recommendation, WWW 2019).

---

> ### Author Response · Authors · 2020-11-16
> **We thank the reviewer for their constructive comments! Please read our rebuttal below.**
>
> We thank the reviewer for mentioning the points about terminology centered around Logic. The mix in terminology is deliberate as we try to cater to both logic and graph communities - semantic web terminologies are more geared towards a reader from the graph community as opposed to someone having prior exposure to the logic background. We hope our terminology helps to be both general as well as easy to understand for the core graph community.
>
> ---
>
> We thank the reviewer for the related works, we will update these works in the paper and will modify our claim to include the previous works on logical reasoning. Specifically, both RuDas and ILP competition are built for rule learning in the scope of prolog/ILP and do not specifically cater to graph neural networks. We will mention these as alternative datasets that practitioners can possibly use for investigating logical understanding in general.
>
> ---
>
> We thank the reviewer for making these interesting observations about our experiments. Due to the scope of our paper to evaluate the generalization capabilities of Graph Neural Networks, we chose not to include baselines that are not of the same family of algorithms. The question here is not whether GNNs are the best models for learning this task. In fact, we agree with the reviewer that GNNs are probably not designed to perform well on such tasks. The question more is how graphs perform this task and what are the strengths and weaknesses of the existing GNNs.
>
> ---
>
> Regarding smaller comments:
>
> - By generalization we expect the model to perform well on held out patterns of the same rule set. By adaptation, we mean the ability of the model to learn a new distribution on new sets of rules.
> - We will define the scope of compositional generalization more clearly in the draft.
> - We make sure that the rule generation process generates sound and consistent rules, as the generation process is stochastic and without proper checks and balances the generated rules can quickly go inconsistent across different worlds.
> - We will move the definition of rule similarity earlier in the main paper.

---

> > ### Comment · AnonReviewer4 · 2020-11-23
> > **Response to author updates**
> >
> > I have read the other reviews and the responses of the authors.
> >
> > I disagree with the authors statement about deliberately mixing terminology. If you use notions from two overlapping vocabularies (e.g., logical "constant" and Semantic Web "relation") which differ in meaning, it is confusing for readers who are not familiar with both logic (e.g., relation=interpretation of a logical predicate) and Semantic Web terminology (e.g., relation=a logical predicate).
> >
> > The authors mention that they plan to update the draft, but I cannot see a revised version, and several reviewers pointed out parts that need to be revised (e.g., about compositional generalization, related work).

---

> > > ### Author Response · Authors · 2020-11-24
> > > **Revision updated**
> > >
> > > Many thanks for your feedback, we have updated the draft with the suggested feedback, which includes changes to the Related Works, clearing the scope of generalization, and incorporating changes in terminologies related to chain-rule.

---

### Official Review · AnonReviewer1 · 2020-10-28
**Proposes a rule-based synthetic graph generator; Biases in the generation process lead to doubts about the paper's conclusions**

**Rating:** 4
**Confidence:** 4

**Review:**

The authors propose a synthetic graph generator to evaluate graph neural networks. The generation process starts with defining rules, subset of rules are used to define a world, each world is then used to sample a graph. Test queries are generated by picking a pair of vertices u, v and generating a path connecting them via the rules. There are some biases in the generation process. For instance, the rules are exclusively open path or chain rules. And as noted above, the test queries mostly stick to a path (the authors allow some variations by adding nodes to vertices already on the path but the path seems to form the backbone of the test query). The remainder of the paper takes a few well known graph neural networks and evaluates them on data generated using GraphLog. Based on these results, the authors claim that E-GAT outperforms RGCNs.

This reviewer acknowledges the need for a benchmark for knowledge base completion. However, GraphLog seems to have one too many biases baked into it for it to form a definitive benchmark. The first bias is the adherence to open path or chain rules. I don't understand the need for this. I can think of two different property nodes p1, p2 hanging off a vertex u leading to an edge with another vertex v. Can this be captured by a chain rule? In section 3, the paper states "Path-based Horn clauses of this form ... encompass the types of logical rules learned by state-of-the-art rule induction systems" and then goes on to cite \partial ILP and NeuralLP. \partial ILP captures more than just chain rules, it tries to capture recursive logic programs, and NeuralLP is based on TensorLog which is much more general than this. If you are going to cite related work then might as well cite it properly. Another bias baked into GraphLog is how it generates its test queries which is mostly a path between two vertices. I found myself wondering whether these biases are what's causing RGCNs to underperform on GraphLog generated data. Such doubts lead me to believe there's some gap in GraphLog that needs bridging. Please don't get me wrong, I'm sure GraphLog will still be used to evaluate KBC approaches but I remain unconvinced that it is general enough to warrant a full conference research paper.

Writing quality wise, the presentation is clear enough with details delegated to the appendices. Related work about graph neural networks and knowledge base completion seems to be well covered. One aspect of related work that seems missing is previously proposed synthetic graph generators. For instance, did the authors try to look for generative approaches for scale-free random graphs? Are these not of interest to the KBC and graph neural networks communities for some reason?

---

> ### Author Response · Authors · 2020-11-16
> **We thank the reviewer for the constructive comments! Please read our rebuttal below.**
>
> We thank the reviewer for raising the point about the bias of chain rules. We imbibe the bias of chain rule due to three reasons:
>
> 1. It allows us to build logically sound resolutions of a given relation by using the facts (edges of the chain) using dyadic Horn Clauses, and
> 2. It allows us to make interpretable model inferences and
> 3. It allows us to control the complexity of the task.
>
> This chain query graph is in no way simple for a model to predict and the test performance varies between 30% to 80%. . To give a concrete example, suppose a chain rule is of the form $r_1(a,b) \land r_2(b,c) \land r_3(c,d) \land r_2(d,e) \rightarrow r_5(a,e)$. This chain rule can be decomposed in many different ways for a learner if they have access to the underlying rules (i.e, decomposing $r_1(a,b) \land r_2(b,c)$ to say $r_x(a,c)$ and so on, please refer to the concrete example posted in response to Reviewer 1 for an idea). Additionally, each graph also consists of an entire subgraph built using the same rules encompassing this query path (which we term as “resolution path”). Thus, this allows the GNN to either determine the relation following dangling paths as the reviewer suggested or “if” the GNN behaves like a recursive learner they can opt for the chain rule path we create, which we use to control the complexity of the task.
>
> ---
>
> Regarding “the authors claim that E-GAT outperforms RGCNs”, we highlight that our focus is not on explicitly ranking GNNs but to understand the strengths and weaknesses of the existing models.
>
> ---
>
> We thank the reviewer for bringing up related work on scale-free random graph generation, and we will add it to our draft.
>
> ---

---

> > ### Comment · AnonReviewer1 · 2020-11-18
> > **More clarity needed**
> >
> > Regarding your reasons for focussing on chain rules: All three reasons would also pertain to the kind of queries answered by approaches such as query2box and betaE. That is, these works are also based upon binary predicates, claim to be interpretable, and describe various queries of increasing complexity. However they do not necessarily stick to chain rules. As examples, see queries 2i, 3i, pi, ip etc. queries in Fig 4 in the query2box paper (by Ren et al in ICLR'20). So I remain a bit unclear on your focus on chain rules.
> >
> > About the comparison between the various approaches considered in the paper: Do you think its possible that your results favoring E-GAT over RGCNs is in part due to the biases in GraphLog? Why or why not?

---

> > > ### Author Response · Authors · 2020-11-18
> > > **Response to reviewer queries**
> > >
> > > We thank the reviewer for their prompt response and the useful follow-up questions:
> > >
> > > 1. Regarding your question about chain queries, we clarify that Graphlog does provide non-chain queries (as a result of subgraph extraction setup). We direct the reviewer to section A2 in the appendix for further details.
> > >
> > > 2. Further, we use only the chain graph to estimate the lower bound on the complexity of a query graph. The model still has access to the non-chain queries generated by the underlying rules.
> > >
> > > 3. Due to the stochastic process of generation, cases such as 2i, 3i, pi, and ip are prevalent in abundance in GraphLog.
> > >
> > > 4. Additionally, GraphLog allows controlling this lower bound on complexity in generation by either increasing the length of the chain graph, or by increasing the number of relations considered, or by increasing the number of rules considered, which dramatically increases the complexity of the task. None of this is explored in query2box.
> > > ----
> > >
> > > GAT has been proven to be superior in performance to RGCN in many works in the literature [1,2], so we do not understand why "biases" in GraphLog would be any more favorable to EGAT.  One potential reason is the attention mechanism could enable or allow the model to ignore or include certain reasoning pathways in the query graph, whereas for RGCN similar capacity is not built inductively in the model.
> > >
> > > [1] https://arxiv.org/abs/1906.01195
> > >
> > > [2] https://arxiv.org/abs/1908.06177

---

### Official Review · AnonReviewer2 · 2020-10-28
**A refinement of synthetic knowledge graph benchmarks**

**Rating:** 6
**Confidence:** 3

**Review:**

This work proposes a method for generating synthetic datasets for testing path-based (knowledge) graph completion. Until recently, there were not many good benchmarks for evaluating reasoning with learned rules or learned knowledge, but there has been a fair amount of work on developing benchmarks for this lately. The synthetic datasets generated here are distinguished by the ability to produce datasets that share a controllable amount of rules. This permits the benchmarks to be used to evaluate multitask learning, robustness to distribution shift, etc. As an illustration of this, the paper includes experiments with a variety of baseline methods showing how (a) the generalization ability of various methods grows and then declines as the number of tasks is increased and (b) fine-tuning on diverse tasks improves accuracy. They also show that in a continual learning setting, the baseline methods all exhibit catastrophic forgetting.

There is value in being able to control the "relatedness" of these synthetic tasks in a principled way (the experiments are a good illustration of how this may be used), so I am leaning towards acceptance. My hesitation is that it's a bit on the incremental side, and seems oversold in a few places, as follows:

The work suggests that the proposed benchmarks examine the "compositional generalization" abilities of models, but it is not clear to me how this is so. The relations are drawn from a fixed set of types. It's true that the rules may examine new combinations of relations, but the end result is just that one of the existing relations is inferred to hold between the start and end of the path. The fact that those rules hold for all bindings is a kind of generalization, but I would not call it "compositional." Compositional generalization would mean something more like learning how to map a grammatical structure to the logical form correctly, irrespective of what was in the constituent phrases.

Also, the work says "GraphLog is the only dataset specifically designed to test logical generalization capabilities on graph data." That seems a little bit of a stretch. CLUTTR seems to feature a very similar set of synthetic benchmarks, the only difference is that CLUTTR *additionally* produces some text, but the abstract focus of the task was an essentially similar knowledge graph completion task. Placing this work side-by-side with CLUTTR, the difference is really (just) in being able to tune the degree of shared structure between multiple datasets. That's valuable, I agree. But this one difference is only discussed by a table featuring four X's for each of the variants of the task setup that this feature enables. The lack of a proper discussion of related work obscures the actual extent of the contribution.

---

> ### Author Response · Authors · 2020-11-16
> **We thank the reviewer for their constructive feedback and for highlighting the strengths of our work. Please read our rebuttal below.**
>
> We thank the reviewer for their question on compositional generalization. We agree with them that the scope of compositional generalization is broad. In this work, we focused on a specific aspect of generalization which involves generalizing to new combinations of existing rules. Since GraphLog is built on top of FOL rules, each graph is constructed by a combination of such rules. By compositional generalization we inspect the ability of a Graph Neural Network to be able to solve unseen combinations of existing rules, thus generalizing through the composition of known concepts. We will make this point clear in the draft.
>
> ---
>
> Regarding comparison with CLUTTR [1], we think CLUTTR is a useful precursor to our work on GraphLog and there are some important differences between the two works:
>
> - CLUTRR consists of a single rule world which in turn contains only 15 rules. GraphLog extends the concept to a more general framework where a user can define how many rules can occur in a single world, as well as define multiple such worlds.
> - GraphLog allows building multiple worlds consisting of either the same, overlapping, or distinct set of rules - which allows practitioners to test multi-task and continual learning scenarios in minute detail by controlling the distribution shift. No such scope is present in CLUTRR, and hence CLUTRR is not suitable for performing Multitask/Continual Learning.
> - CLUTRR is built on a static set of relations (22 family relations) while GraphLog can contain any number of such relations since it's a flexible generator along with a dataset.
>
> Regarding “difference is really (just) in being able to tune the degree of shared structure between multiple datasets”, we highlight that enabling control over the degree of shared structures is not a trivial exercise and needs rigorous empirical evaluation to ensure that changing the degree of shared structures actually leads to meaningful change in the model’s performance.
>
> We thank the reviewer for their suggestion and would update the related work to clearly outline the differences between CLUTTR and our work.
>
> [1] Sinha, Koustuv, et al. "Clutrr: A diagnostic benchmark for inductive reasoning from text." arXiv preprint arXiv:1908.06177 (2019).

---

### Official Review · AnonReviewer3 · 2020-10-29
**An interesting dataset but efficacy not demonstrated**

**Rating:** 5
**Confidence:** 4

**Review:**

I liked the idea of GraphLog; it seems like an interesting and potentially very useful dataset. The "results" seemed little to do with the dataset (apart from being enabled by the dataset), but were comparing some models that were neither the current state-of-the-art or particularly novel. I could imagine a paper that does a comprehensive reevaluation the state-of-the-art algorithms (unchanged!) using the new data set and then reports on which ones work well for different tasks. Such reproducibility in science is important. However, this is not was done.

I would like to see more discussion about why doing well on the benchmark would translate into doing well on real-world problems. I would hypothesize that the real-world does not consist of a few simple rules chained together, but the underlying dynamics is much more complicated, and agents are trying to concoct a story to get some signal out of the noise. All of your tests assume there is a clean signal -- constructed by your rules (perhaps corrupted by random noise(?)).  You just assume that this will form a good testbed, but it isn't obvious to me.

The rules all seem overly simplistic. I was expecting your rule generator to generate more complex rules from simpler ones. But it doesn't; they all seem to be just the same complexity.  Here are some things I'm wondering:   By sharing subset of rules, the worlds are structurally similar? Are the entities shared in any meaningful way? I think they are just meaningless names, and so there is no commonalities between the worlds. I think they share relations. Is there any notion that a relation learned in one world are like the relation with the same name in other worlds? You are not learning about relations using diverse populations, or are you?

Minor comments

In RGCN, $\times_i$ what is $i$?

Figure 3, I thought was weird. If we divide the datasets into easy medium and hard, don't we always have a nice straight lines when plotted with difficulty? I think it is just trying to say that the E-GAT models work better across difficulties.

In figure 5, I can't see how/why "The colors of the bars depict how similar two distributions are"



After rebuttal:


Comment:
I am suspicious of your argument on "complex due to multiple resolution pathways".

Surely there is always a (unique) canonical deviation. For example, one could imagine extending Prolog to allow for any literal to be resolved; not just the left one. However that just adds extra complexity. Because we have to resolve all literals, we might as well do in a left-to-right order. Surely in your case, we can always do the leftmost (for example) one. Or the rules can be defined so that we can resolve them left to right.

One counter to my argument might be (parent, ancestor) -> ancestor parent -> ancestor.

Here we cannot go left-to-right, but need to go right to left. We do not need to search over orderings.

This issue is faced daily by Prolog programmers; we choose whichever one of (using your notation): (parent, ancestor) -> ancestor (ancestor, parent) -> ancestor works with your engine. I don't see why they get around it without problem and you claim it is not possible. The rules may depend on the order used, but what is the problem with that if it drastically reduces the search space? You don't have to search for all proofs; just one.

---

> ### Author Response · Authors · 2020-11-16
> **Thanks to the reviewer for constructive comments! Please read our rebuttal below [1/2]**
>
> [1/2]
>
> ---
>
> We thank the reviewer for their constructive feedback and we are glad that they find GraphLog “interesting” and “potentially very useful”. The reviewer raises an important question that our baselines are neither SOTA nor novel. We highlight that most of the SOTA GNN models are not designed for relational data. Even for models like GAT, we used the EGAT variants proposed in the literature by [1]. Additionally, we are working on adding the results with the GIN model. Regarding models not being novel, our objective was not to develop a new model but highlight the strengths and weaknesses of the existing models. We respectfully disagree with the reviewer that “The results seemed little to do with the dataset”. Our “dataset” provides a mechanism to control data distribution shift, thus enables quantifying the performance of the different GNN models on the task of logical relation reasoning, in three setups: (i) Single Task Learning, (ii) Multi-Task Training, and (iii) Continual Learning. To marginalize the contribution of the proposed “dataset” by saying that they just enable the results is similar to marginalizing the contribution of a proposed “model” by saying that they just enable the results.
>
> [1] Sinha, Koustuv, et al. "Clutrr: A diagnostic benchmark for inductive reasoning from text." arXiv preprint arXiv:1908.06177 (2019).
>
> -----
>
> The reviewer raises an interesting point and we agree that we do not provide any evidence of why doing well on the benchmark would translate to doing well on the real-world tasks. However, the benchmark enables evaluating the models for compositionality, which can not be easily tested for real-world datasets. The point of the benchmark is not to be an indicator of success on real-world tasks, but to provide a diagnostic setup for properties that are not easy to evaluate in existing real-world datasets and that are known to be important for human-level abstract reasoning and planning.
>
> -----
>
> We argue the underlying dynamics of GraphLog is very complex, and would require a GNN to perform compositional reasoning. To give a concrete example, a noise-free graph in our setup can be as follows:
>
> $a \xrightarrow[]{r_1} b \xrightarrow[]{r_2} c \xrightarrow[]{r_3} d \xrightarrow[]{r_2} e$ ,
>
> where, a logical resolution of this graph could yield the relation $r_5$ : $r_1(a,b) \land r_2(b,c) \land r_3(c,d) \land r_2(d,e) \rightarrow r_5(a,e)$
>
> where $r_5$ is the target relation to predict between the nodes $a$ and $e$, which is not available to the graph. (Note that our graphs are rarely noise-free, which we elaborate below). Say we have the following underlying rules:
>
> 1. ${(r_1, r_2) \rightarrow r_3}$,
> 2. ${(r_3, r_2) \rightarrow r_4}$,
> 3. ${(r_3, r_4) \rightarrow r_5}$,
> 4. ${(r_2, r_3) \rightarrow r_6}$,
> 5. ${(r_6, r_2) \rightarrow r_4}$,
> 6. ${(r_1, r_4) \rightarrow r_5}$
>
>
> If a learner has access to the true underlying rules then there can exist two different ways to resolve the same query:
>
> Execution 1:
> 1. $r_1(a,b) \land r_2(b,c) \land r_3(c,d) \land r_2(d,e)$ → use 1
> 2. $r_3(a,c) \land r_3(c,d) \land r_2(d,e)$ → use 2
> 3. $r_3(a,c) \land r_4(c, e)$ → use 3
> 4. $r_5(a, e)$
>
> Execution 2:
> 1. $r_1(a,b) \land r_2(b,c) \land r_3(c,d) \land r_2(d,e)$  → use 4
> 2. $r_1(a,b) \land r_6(b,d) \land r_2(d,e)$ → use 5
> 3. $r_1(a,b) \land r_4(b, e)$ → use 6
> 4. $r_5(a,e)$
>
> - Now even for a rule-based learner having access to the underlying rules, the task is complex due to multiple resolution pathways, and each having many degenerate solutions by rule application. For a GNN, this becomes even more complex due to the absence of rules. Still, a GNN can just potentially memorize all possible edge combinations for relations. To counter that, we focus on inductive reasoning, where a GNN is shown a novel chain path during testing, forcing the model to reason in a compositional manner.
> - Furthermore, we add a subgraph around the resolution chain which is built using the same rules. This subgraph while not potentially useful for the logical resolution of the task can aid or inhibit the reasoning process of GNN. We take extra care of removing any shorter length path than the query path, removing the probability of the model to memorize shortcut connections.
> - As we show above with a trivial example, even with dyadic Horn clauses (or rules) it is possible to generate complex reasoning chains that vary in complexity due to the length of the chain as well as the number of ways the learner can resolve a chain. Thus, we respectfully disagree that the problem formulation is “overly simplistic” or have the “same complexity”.
>
> We sample the rules procedurally, thus the complexity of the graphs generated across different worlds varies drastically based on the closure of the rules generated. We treat entities as a cloze task form, where across graphs there is no sharing of entity representation.

---

> > ### Author Response · Authors · 2020-11-16
> > **Rebuttal continued [2/2]**
> >
> > [2/2]
> >
> > “$i$” denotes one of the modes (dimensions) of the tensor. In the paper, we will update the notation to use “$j$” (and describe what this is) so that the reader does not confuse this with the relation index.
> >
> > ---
> >
> > While the reviewer is absolutely correct that if we divide the datasets into easy medium and hard, we will have a nice line, do note that the difficult splits are not handed over to use by someone. When we generate the datasets, we can control the complexity to only some extent. We work with the intuition that certain design choices will make the dataset harder etc. However, we need to perform experiments to ensure that our intuition is valid and the resulting datasets are indeed hard. Regarding plotting different models, we believe it is important to be able to make a general claim that dataset X is easier than dataset Y. In absence of multiple models, we can not conclude if data X is easier for model A or easier in general.
> >
> > ---
> >
> > Thank you for the question about the color bars. We note that the statement: “The colors of the bars depict how similar two distributions are” is not an inference from the results. We use different colors for the bars when making the plots, to show how similar they are. The point we want to highlight is, more similar are the datasets, the better is the few shot generalization performance.

---

> > ### Comment · AnonReviewer3 · 2020-11-20
> > **Your argument on complexity**
> >
> > I am suspicious of your argument on "complex due to multiple resolution pathways".
> >
> > Surely there is always a (unique) canonical deviation. For example, one could imagine extending Prolog to allow for any literal to be resolved; not just the left one.  However that just adds extra complexity. Because we have to resolve all literals, we might as well do in a left-to-right order.  Surely in your case, we can always do the leftmost (for example) one. Or the rules can be defined so that we can resolve them left to right.
> >
> > One counter to my argument might be
> > (parent, ancestor) -> ancestor
> > parent -> ancestor.
> >
> > Here we cannot go left-to-right, but need to go right to left.  We do not need to search over orderings.
> >
> > This issue is faced daily by Prolog programmers; we choose whichever one of (using your notation):
> > (parent, ancestor) -> ancestor
> > (ancestor, parent) -> ancestor
> > works with your engine. I don't see why they get around it without problem and you claim it is not possible. The rules may depend on the order used, but what is the problem with that if it drastically reduces the search space?  You don't have to search for all proofs; just one.

---

### Author Response · Authors · 2020-11-24
**Paper updated with addressing reviewer concerns**

We have updated the paper following the concerns from reviewers. The majority of the changes were done to address the following:

- Related Works - incorporated feedback from R2, R1, and R4
  - Defined scope of compositional generalization
  - Added a detailed comparison with CLUTRR.
  - Added a section on Synthetic Graph Generation.
- Terminologies - incorporated feedback from R3, R1, and R4
  - Clarified the equation to being a formulation of chain-rule
  - Added section on Controlling similarity in the main paper

---

### Decision · Program_Chairs · 2021-01-07
**Final Decision**

**Decision:**

Reject

**Comment:**

The reviewers point out several important issues to be addressed, including comparing to other methods that can address the "combinatorial generalization" problems studied (one reviewer points out the crucial difference from "compositional generalization" studied before), addressing the gap between the proposed dataset (simple and has the value of diagnosing/model debug/research algorithm development) and real datasets/problem settings.

As such the AC recommends Reject and encourages the authors to take the constructive feedback to improve.